# SOIL: Contrastive Second-Order Interest Learning for Multimodal Recommendation

## ABSTRACT

Mainstream multimodal recommender systems are designed to learn user interest by analyzing user-item interaction graphs. However, what they learn about user interest needs to be completed because historical interactions only record items that best match user interest (i.e., the first-order interest), while suboptimal items are absent. To fully exploit user interest, we propose a **S**econd-**O**rder **I**nterest **L**earning (**SOIL**) framework to retrieve second-order interest from unrecorded suboptimal items. In this framework, we build a user-item interaction graph augmented by second-order interest, an interest-aware item-item graph for the visual modality, and a similar graph for the textual modality. In our work, all three graphs are constructed from user-item interaction records and multimodal feature similarity. Similarly to other graph-based approaches, we apply graph convolutional networks to each of the three graphs to learn representations of users and items. To improve the exploitation of both first-order and second-order interest, we optimize the model by implementing contrastive learning modules for user and item representations at both the user-item and item-item levels. The proposed framework is evaluated on three real-world public datasets in online shopping scenarios. Experimental results verify that our method is able to significantly improve prediction performance. For instance, our method outperforms the previous state-of-the-art method MGCN by an average of 8.1% in terms of Recall@10.

## CCS CONCEPTS

• **Information systems** → **Recommender systems**; **Multimedia and multimodal retrieval**.

## KEYWORDS

Multimodal Recommendation, Second-Order Interest, Attractiveness Score, Contrastive Learning

## 1 INTRODUCTION

In recent years, mainstream online service providers have offered online services that integrate multimedia information (e.g., images, texts, and videos) and have established multimodal recommender systems. A typical multimodal recommender system consists of a group of users, a series of items, and the corresponding multimedia data for each item. Service providers design various solutions

*ACM MM, 2024, Melbourne, Australia*
© 2024 Copyright held by the owner/author(s). Publication rights licensed to ACM.
ACM ISBN 978-x-xxxx-xxxx-x/YY/MM
https://doi.org/10.1145/nnnnnnn.nnnnnnn

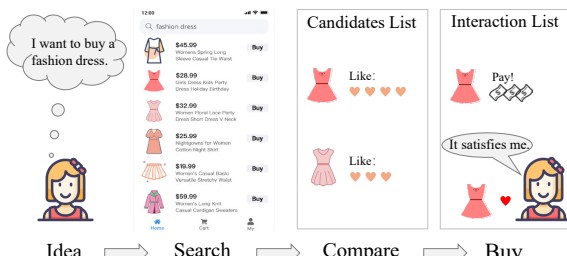

**Figure 1: Illustration of user behaviors in online shopping. Mainstream multimodal methods only use items in the interaction list to represent user interest, yet items in the candidate list are also an essential part of user interest.**

to predict possible user-item pairs based on historical user-item interactions [23, 29].

Among these multimodal recommendation solutions, most of them are graph-based approaches that utilize graph neural networks to exploit the relationship between user-item interactions and multimodal item embeddings [4, 18, 20, 23, 24, 26, 28]. Graph-based approaches generally begin by constructing an interaction-based user-item graph and an item-item relationship graph, and then perform graph convolutional learning on these graphs separately. Finally, the outputs of two graph convolutional networks are fused to predict the user-item matching pairs. Experimental results in recent reports have demonstrated that learning on interaction-based graphs improves prediction accuracy significantly.

Despite the great success of interaction-based graph learning methods, more attention must be paid to the fact that relying solely on historical user-item interaction records is insufficient to represent user interest. For better understanding, we illustrate a typical sequence of user behaviors in the online shopping scenario in Figure 1. As illustrated in Figure 1, a person who is looking for fashionable spring clothing will first choose the candidate items that attract her and then click or buy the item that best matches her interest (In this paper, we refer to this phenomenon as the **Best-Match Trading Principle**.). The user behavior sequence indicates that items in the candidate list are part of user interest and that the user is likely to select these items in the future. For example, when an item that we have previously purchased is out of stock, we tend to select one from the candidate list for purchase. The experimental results in Figure 6 also verify that users do purchase the items in the candidate list reconstructed in this paper. Recommender systems that only use items in the interaction list are agnostic about the items in the candidate list, thus degrading recommendation performance. Taking into account this phenomenon, a recommender system should simultaneously manage the items on the interaction and candidate lists to thoroughly understand user interest. However, most previous studies failed to utilize these candidate items because they were not recorded in interaction logs.

To fully exploit user interest, we aim to identify potential candidate items for each user utilizing multimodal information. By

investigating previous studies, we outline three critical issues in mainstream multimodal recommendation solutions:

(1) **Incomplete User-Item Graph:** The absence of interaction information between users and candidate items in the user-item graph leads to insufficient learning of user interest and thus degrades the recommendation performance.

(2) **Interest-agnostic Item-Item Graph:** The absence of user interest information in the item-item graph leads to a vulnerable relationship between users and multimodal information, deteriorating recommendation performance.

(3) **Limited Interest Consistency:** Inconsistency in user interest between multimodal embeddings and user-item embeddings for the same interaction record will lead to low ranking weights and degrades recommendation performance.

To challenge these issues, we propose a solution to retrieve potential candidate items for each user and optimize the recommendation model with contrastive learning techniques, named the **S**econd-**O**rder **I**nterest **L**earning (**SOIL**) framework. **In this paper, for each user, we denote all items on the interaction list as first-order interest and all items on the candidate list as second-order interest.** In the proposed SOIL framework, we explicitly construct a second-order user-item interaction graph and interest-aware item-item graphs, and propose to enhance interest consistency with a user-item level contrastive learning module and an item-item level contrastive learning module. Technically, **to handle the challenge of incomplete user-item graph**, we first calculate the attraction score for each potential user-item pair based on the similarity of multimodal features, and then construct the augmented user-item interaction graph that incorporates both attraction scores and interaction logs. **To handle the challenge of interest-agnostic item-item graph**, we first compute interest representations for each user through multimodal features, and then construct an interest-aware item-item graph based on the similarity between multimodal features and interest representations. **To handle the challenge of limited interest consistency**, we first design a contrastive learning module at the user-item level to enhance user interest for each user in a self-supervised learning manner. Furthermore, we design an item-item level contrastive learning module to maintain the consistency of user interest between user-item embeddings and multimodal embeddings.

In summary, the contributions of our work are as follows:

(1) We identify that due to the best-match trading principle, user-item interactions contain only the items that best match the user interest, while suboptimal items are absent. To mitigate this problem, we propose a **S**econd-**O**rder **I**nterest **L**earning (**SOIL**) framework to exploit the second-order interest from unrecorded suboptimal items.

(2) We propose to augment the user-item graph with second-order interest and construct interest-aware item-item graphs through user interest. With these graphs, we conduct contrastive learning at both the user-item and item-item levels to enhance user interest.

(3) We conduct extensive experiments on three public datasets in online shopping scenarios. Experimental results demonstrate that our method is able to significantly outperform state-of-the-art methods by an average of 6.8% in terms of Recall@10.

## 2 RELATED WORK

### 2.1 Multimodal Recommendation

Multimodal recommendation models utilize multimedia information (e.g., images, texts, and videos) and user-item interactions to provide personalized recommendation services [2, 21, 24, 25]. Traditional multimodal recommendation methods mainly incorporate multimodal information into collaborative filtering [15] or matrix factorization [12] models to improve prediction performance [6, 11]. Among them, VBPR [6] is a method based on matrix factorization that employs pre-trained convolutional neural networks to extract visual features, which are then integrated into item embeddings. MAML [10] is designed to exploit user preferences with an attention network. This method first estimates the user attention for each item and then incorporates the attention information into matrix factorization models. Inspired by graph convolution networks (GCNs) [9], researchers have proposed various graph-based methods to learn user preferences [18, 20, 21, 24, 29]. Among them, MMGCN [21] is a representative method that constructs a bipartite user-item graph for each modality to implicitly exploit the multimodal item relationship. To directly exploit the item-item relationship, Zhang et al. [24] specifically designed an item-item graph to utilize multimodal features. Recent studies [4, 23, 26, 28] have verified that explicitly constructing item-item graphs significantly improves recommendation performance. Among them, FREEDOM [28] demonstrates that dynamic item-item graphs will degrade recommendation performance and presents a denoising framework with a fixed item-item graph.

### 2.2 Graph-based Recommendation

Graph-based recommendation models treat the information in recommender systems as various graphs and leverage graph convolution networks to embed user-item interactions [22]. LightGCN [7] is a representative graph-based recommendation method that simplifies the vanilla graph convolution network to satisfy the requirements of recommender systems. In multimodal recommender systems, graph-based recommendation methods have demonstrated extensive ways to construct graphs. For instance, GRCN [20] constructs a dynamic interaction graph and adaptively refines the structure of the interaction graph based on the training status. DualGNN [18] constructs a bipartite user-item graph and a user co-occurrence graph to exploit dynamic user preferences. LAT-TICE [24] presents a modality-aware structure learning network to exploit the item relationship for each modality and constructs an item-item graph based on the learned item relationship. MGCN [23] constructs an item-item graph for each modality and performs graph convolution learning on these graphs separately. Similarly, DRAGON [26] first constructs an item-item graph for each modality and then fuses these modality graphs into a multimodal item-item graph. LGMRec [4] learns hypergraphs for each modality to exploit the dependency relationship between items.

Our work aims to build graphs that incorporate interaction records and multimodal features to capture user interest in a comprehensive way. To this end, we construct a user-item graph that contains both first-order and second-order interest. Furthermore, we propose to incorporate user interest into the item-item graph to leverage user preferences across various modalities.

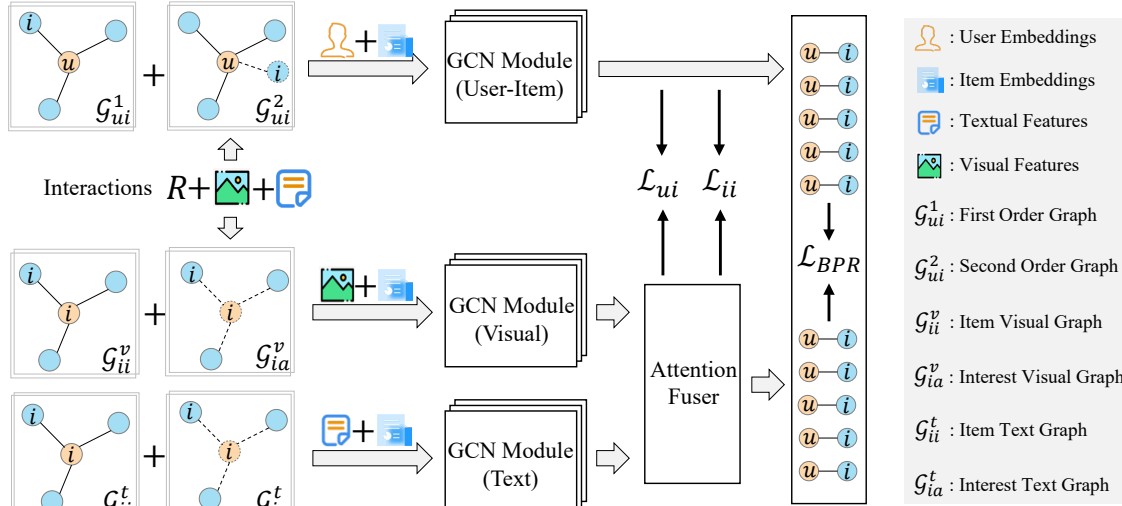

**Figure 2: Illustration of the Second-Order Interest Learning framework. The proposed framework consists of three distinct graphs: a user-item interaction graph augmented by second-order interest, an interest-aware item-item relationship graph for the visual modality, and a similar graph for the textual modality. In this framework, graph convolutional networks are initially deployed on each of the three graphs to learn representations of users and items. Subsequently, contrastive learning is applied at both the user-item and item-item levels to effectively capture user interest.**

## 3 PROPOSED METHOD

### 3.1 Problem Formulation

A typical multimodal recommendation system generally consists of a group of users, a series of items, and corresponding multimedia data for each item. In our work, we denote the set of $m$ users by $\mathbf{U} = \{u_1, u_2, ..., u_m\}$, and the set of $n$ items by $\mathbf{I} = \{i_1, i_2, ..., i_n\}$. We denote the multimodal data of $n$ items by $\mathbf{M} = \{M_1, M_2, ..., M_n\}$, where $M_i = \{M_i^v, M_i^t\}$ refers to the visual and textual features of the $i$-th item. The interaction matrix $\mathbf{R} \in \{0, 1\}^{|\mathbf{U}| \times |\mathbf{I}|}$ is a collection of all user-item interactions, where $r_{u,i} \in \{0, 1\}$ indicates whether user $u$ interacts with item $i$. Note that $\mathbf{R}$ is also referred to as the first-order interaction matrix in this paper.

For better understanding, we illustrate the proposed Second-Order Interest Learning (SOIL) framework in Figure 2. In this framework, we construct an augmented user-item interaction graph that contains both first-order and second-order interest, an interest-aware item-item relationship graph for the visual modality, and an interest-aware item-item graph for the textual modality. The three graphs are built using user-item interaction records and multimodal features, and the process of constructing these graphs will be detailed in the subsequent sections. With these three graphs, graph convolutional networks can be implemented to learn the representations of users and items. To enhance the exploitation of user interest and maintain consistency in user interest across the three graphs, we implement contrastive learning modules at both the user-item and item-item levels for user and item representations. Finally, we optimize the whole model with the pairwise Bayesian personalized ranking loss.

### 3.2 Second-Order Interaction Graph

As illustrated in Figure 1, we identify that the items on the candidate list are not recorded. However, these candidates are crucial aspects of user interest that can improve recommendation performance. In this paper, we denote all candidate items of a user as the second-order interest of that user. To retrieve candidate items and exploit the second-order interest, we first compute the attractiveness score for each potential user-item pair and then append potential user-item pairs to the interaction graph by taking attractiveness scores as edge weights. The augmented interaction graph is depicted in Figure 3, with dashed lines representing the newly added edges and the weights corresponding to the scores of the items that attract users. For instance, path "$u_2$- -$i_3$" indicates that item $i_3$ has an attractiveness score of 0.85 for user $u_2$.

In our work, the attractiveness score is calculated by the similarity between an item and the first-order interest of a user. We first calculate the similarity of the features for each modality to obtain attractiveness scores. The similarity matrix $S \in \mathbb{R}^{n \times n}$ consists of $n \times n$ similarity scores, where the element $s_{i,j}$ in row $i$ and column $j$ is calculated with a cosine similarity function as follows:

$$s_{i,j} = \frac{M_i \cdot M_j^{\mathrm{T}}}{\|M_i\|\|M_j\|}, \tag{1}$$

where $M_i$ and $M_j$ refer to the multimodal features of item $i$ and item $j$. In our work, we calculate the similarities of visual and textual features separately and denote them as $s_{i,j}^v$ and $s_{i,j}^t$.

Based on Eq.(1), we calculate feature similarity matrices $S^v$ and $S^t$ for visual and textual modalities. In our work, the attractiveness score matrix is calculated by the dot product of $S^v$ and $S^t$, denoted as $S^a = S^v \cdot S^t$. Each element $s_{i,j}^a$ in the matrix $S^a$ represents the similarity between the item $i$ and the item $j$. With matrix $S^a$, we select items similar to the first-order interest as the second-order interest for each user. The process of selecting second-order interest for user $u$ is formulated as follows:

$$SOI\text{-}W(u,i), SOI\text{-}I(u) = \text{top-}k\left(\left\{S_{i,:}^a | r_{u,i} = 1, r_{u,i} \in \mathbf{R}\right\}\right), \tag{2}$$

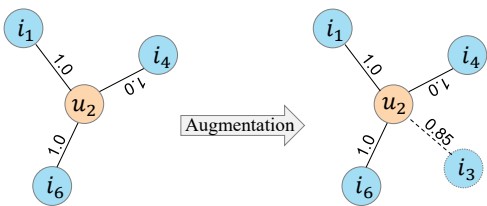

**Figure 3: Illustration of the second-order interaction graph.**

where $SOI\text{-}I(u)$ refers to the set of selected items for user $u$, and $SOI\text{-}W(u,i)$ records the attractiveness score for each $i \in SOI\text{-}I(u)$, top-$k(\cdot)$ refers to the function to select the maximum $k$ elements, $S_{i,:}^a$ refers to the $i$-th row in attractiveness matrix, $r_{u,i} = 1$ indicates that item $i$ is in the first-order interest of user $u$. This equation shows that the second-order interest is made up of $k$ items that are most similar to the first-order interest.

With Eq.(2), we are able to build the second-order interaction matrix $\mathbf{R}^a$ as follows:

$$\mathbf{R}_{u,i}^a = \begin{cases} SOI\text{-}W(u,i), & i \in SOI\text{-}I(u), \\ 0, & \text{otherwise.} \end{cases} \tag{3}$$

The second-order interaction matrix $\mathbf{R}^a$ shares the same shape with the first-order interaction matrix $\mathbf{R}$, allowing them to be integrated into the user-item interaction graph to learn collaborative representations. In our work, we implement a light graph convolutional network (lightGCN) [7] to compute collaborative information on the user-item interaction graph. The computation of the $l$-th layer is formulated as follows:

$$\mathbf{E}_{ui}^l = (\mathbf{L})\mathbf{E}_{ui}^{l-1}, \tag{4}$$

where $\mathbf{E}_{ui}^l$ refers to the embedding matrix of the $l$-th layer. For the first layer, the initial embedding matrix $E_{ui}^0 = E_u \oplus E_i$ is the concatenation of the user embedding matrix $E_u$ and the item embedding matrix $E_i$. $\mathbf{L}$ refers to the sum of the Laplacian matrix for the first-order interaction graph and the Laplacian matrix for the second-order interaction graph. It is formulated as follows:

$$\mathbf{L} = \mathbf{D_1}^{-1/2}\mathbf{A_1}\mathbf{D_1}^{-1/2} + \mathbf{D_2}^{-1/2}\mathbf{A_2}\mathbf{D_2}^{-1/2},$$
$$\mathbf{A_1} = \begin{vmatrix} 0 & \mathbf{R} \\ \mathbf{R}^{\mathrm{T}} & 0 \end{vmatrix}, \quad \mathbf{A_2} = \begin{vmatrix} 0 & \mathbf{R}^a \\ (\mathbf{R}^a)^{\mathrm{T}} & 0 \end{vmatrix}, \tag{5}$$

where $\mathbf{D_1}$ is the diagonal degree matrix of $\mathbf{R}$, whose $i$-th element is the sum of the $i$-th row in $\mathbf{R}$. Similarly, $\mathbf{D_2}$ is the diagonal degree matrix of $\mathbf{R}^a$. Note that we compute the Laplacian matrices of the two graphs separately to maintain the original structures inside the two graphs.

### 3.3 Interest-aware Item-Item Graph

According to Section 2.2, previous multimodal recommendation approaches have constructed various item-item graphs based on feature similarity and $k$NN sparsification [1, 23, 24, 28]. However, these graphs are agnostic to interest and are not able to learn user interest to improve recommendation performance. To enhance the relationship between multimodal features and users, we propose to create interest-aware item-item graphs for each modality.

In our work, we propose to construct interest-aware graphs with feature similarity matrices $S^v$ and $S^t$, which are calculated with Eq.(1). With $S^v$ and $S^t$, we denote user interest by the sum of feature similarities, which is formulated as follows:

$$\begin{aligned} S^v(u) = \sum S_{i,:}^v, & \quad i \in I(u), \\ S^t(u) = \sum S_{i,:}^t, & \quad i \in I(u), \end{aligned} \tag{6}$$

where $I(u) = \{i|r_{u,i} = 1, r_{u,i} \in \mathbf{R}\}$ refers to the set of items that user $u$ has interacted with, $S^v(u)$ refers to the sum of visual similarity of all items in $I(u)$, and $S^v(u)$ has the shape of $1 \times n$. $S^t(u)$ refers to the sum of textual similarity. With $S^v(u)$ and $S^t(u)$, we are able to select top-$k$ items to represent user interest. The process of selecting top-$k$ items is formulated as follows:

$$I^v(u) = \text{top-}k\left(S^v(u)\right), \quad I^t(u) = \text{top-}k\left(S^t(u)\right), \tag{7}$$

and we denote the concatenation set of $I^v(u)$ and $I^t(u)$ by $I^a(u) = I^v(u) \cup I^t(u)$.

With top-$k$ items representing user interest, we construct the interest-aware graphs as follows:

$$\begin{aligned} \hat{S}_{i,j}^v &= \begin{cases} S_{i,j}^v, & i \in I(u) \text{ and } j \in I^a(u), \\ 0, & \text{otherwise.} \end{cases} \\ \hat{S}_{i,j}^t &= \begin{cases} S_{i,j}^t, & i \in I(u) \text{ and } j \in I^a(u), \\ 0, & \text{otherwise,} \end{cases} \end{aligned} \tag{8}$$

where $\hat{S}_{i,j}^v$ and $\hat{S}_{i,j}^t$ refer to the interest-aware graphs for visual and textual modalities, respectively. To maintain the latent relationship among multimodal features, we conduct $k$NN sparsification [1] on $S^v$ and $S^t$ to construct similarity graphs as follows:

$$\begin{aligned} \tilde{S}_{i,j}^v &= \begin{cases} S_{i,j}^v, & j \in \text{top-}k(S_{i,:}^v), \\ 0, & \text{otherwise.} \end{cases} \\ \tilde{S}_{i,j}^t &= \begin{cases} S_{i,j}^t, & j \in \text{top-}k(S_{i,:}^t), \\ 0, & \text{otherwise.} \end{cases} \end{aligned} \tag{9}$$

In our work, we normalize $\hat{S}_{i,j}^v$, $\hat{S}_{i,j}^t$, $\tilde{S}_{i,j}^v$ and $\tilde{S}_{i,j}^t$ in the same way to guarantee robust graph convolutional learning. Taking $\hat{S}_{i,j}^v$ as an example, the normalization process is expressed as $\hat{S}_{i,j}^v = (D)^{-\frac{1}{2}}\hat{S}_{i,j}^v(D)^{-\frac{1}{2}}$, where $D_{ii} = \sum_j \hat{S}_{i,j}^v$ is the diagonal degree matrix of $\hat{S}_{i,j}^v$. Once the normalization is done, we merge the interest-aware graphs and similarity graphs for each modality into the interest-aware item-item graphs as follows:

$$\mathcal{G}^v = \hat{S}_{i,j}^v + \tilde{S}_{i,j}^v, \quad \mathcal{G}^t = \hat{S}_{i,j}^t + \tilde{S}_{i,j}^t. \tag{10}$$

To learn interest-aware multimodal representations, we implement graph convolutional networks on $\mathcal{G}^v$ and $\mathcal{G}^t$, and formulate the computation of the $l$-th layer as:

$$\mathbf{E}_{iv}^l = (\mathcal{G}^v)\mathbf{E}_{iv}^{l-1}, \quad \mathbf{E}_{it}^l = (\mathcal{G}^t)\mathbf{E}_{it}^{l-1}, \tag{11}$$

where $\mathbf{E}_{iv}^l$ and $\mathbf{E}_{iv}^l$ refer to the embedding matrices of the $l$-th layer for visual and textual modalities. For the first layer, initial embedding matrices $E_{iv}^0$ and $E_{it}^0$ are calculated by fusing multimodal features with item embeddings as follows:

$$E_{iv}^0 = E_i \cdot MAP\text{-}V(\mathbf{M}^v), \quad E_{it}^0 = E_i \cdot MAP\text{-}T(\mathbf{M}^t), \tag{12}$$

where $MAP\text{-}V(\cdot)$ and $MAP\text{-}T(\cdot)$ are linear mapping functions or networks for visual and textual features, $\mathbf{M}^v$ and $\mathbf{M}^t$ are multimodal features for all items in visual and textual modalities.

Based on interest-aware multimodal representations $E_{iv}$ and $E_{it}$ derived from Eq.(11), we integrate them with interaction graphs to enhance user interest in multimodal features. The process of integration is formulated as follows:

$$\tilde{E}_{ui}^v = (\mathcal{G}_{ui}E_{iv}) \oplus E_{iv}, \quad \tilde{E}_{ui}^t = (\mathcal{G}_{ui}E_{it}) \oplus E_{it}, \qquad (13)$$

where $\mathcal{G}_{ui} = \mathbf{R} + \mathbf{R}^a$ refers to the interaction graph that contains both first-order and second-order interest, $\oplus$ refers to the concatenate operation. Finally, we fuse multimodal representations of all modalities with an attention mechanism [17, 19, 23, 26], which is formulated as follows:

$$\tilde{E}_{ui} = \tilde{E}_{ui}^v + \tilde{E}_{ui}^t - (\alpha \cdot \tilde{E}_{ui}^v + (1-\alpha) \cdot \tilde{E}_{ui}^t), \qquad (14)$$

where $\alpha = \text{softmax}(Q(\tilde{E}_{ui}^v) \oplus Q(\tilde{E}_{ui}^v))$ refers to the attention weights, and $Q(\cdot)$ refers to the widely used linear attention layer [17, 23].

### 3.4 Pairwise Contrastive Optimization

As mentioned in Section 1, the inconsistency of user interest between user-item embeddings and multimodal embeddings seriously degrades recommendation performance. To handle this challenge, we propose to first enhance user interest in multimodal features and then deploy an interest consistency maintenance module. To enhance the user interest, we design a contrastive learning module on embeddings of users and items as follows:

$$\begin{aligned}
\mathcal{L}_{ui} = &\sum_{r_{u,i}=1, r_{u,i} \in \mathbf{R}} -\log \frac{\exp(\hat{e}_u \cdot e_i / \tau)}{\sum_{r_{\hat{u},i} \in \mathcal{R}} \exp(\hat{e}_u \cdot e_{\hat{i}} / \tau)} \\
&+ \sum_{r_{u,i}=1, r_{u,i} \in \mathbf{R}} -\log \frac{\exp(\hat{e}_u \cdot \tilde{e}_i / \tau)}{\sum_{r_{\hat{u},i} \in \mathcal{R}} \exp(\hat{e}_u \cdot \tilde{e}_{\hat{i}} / \tau)},
\end{aligned} \qquad (15)$$

where $\hat{e}_u$ is user representation derived from embedding matrix $E = E_{ui} + \tilde{E}_{ui}$, $e_i$ and $\tilde{e}_i$ are item representations derived from embedding matrices $E_{ui}$ and $\tilde{E}_{ui}$, and $\tau$ is the temperature factor.

To maintain the consistency of user interest between user-item embeddings and multimodal embeddings, we propose to deploy a contrastive learning module on embeddings of user-item graph and item-item graph as follows:

$$\begin{aligned}
\mathcal{L}_{ii} = &\sum_{i \in \mathbf{I}} -\log \frac{\exp(e_i \cdot \tilde{e}_i / \tau)}{\sum_{\hat{i} \in \mathbf{I}} \exp(e_{\hat{i}} \cdot \tilde{e}_{\hat{i}} / \tau)} \\
&+ \sum_{u \in \mathbf{U}} -\log \frac{\exp(e_u \cdot \tilde{e}_u / \tau)}{\sum_{\hat{u} \in \mathbf{U}} \exp(e_{\hat{u}} \cdot \tilde{e}_{\hat{u}} / \tau)},
\end{aligned} \qquad (16)$$

where $e_i$ and $\tilde{e}_i$ are item representations derived from embedding matrices $E_{ui}$ and $\tilde{E}_{ui}$, $e_u$ and $\tilde{e}_u$ are user representations, and $\tau$ is the temperature factor.

### 3.5 Overall Optimization Strategy

In our work, we first construct a second-order interaction graph to learn second-order interest, and then construct interest-aware item-item graphs for visual and textual modalities. We perform graph convolutional learning on each of the three graphs and obtain a embedding matrix $E_{ui}$ for the user-item graph, and an embedding matrix $\tilde{E}_{ui}$ for the item-item graph. Following the widely used

**Table 1: Statistics of evaluation datasets.**

| Dataset | # Interactions | # Users | # Items | Sparsity |
|---|---|---|---|---|
| Baby | 160,792 | 19,445 | 7,050 | 99.88% |
| Sports | 296,337 | 35,598 | 18,357 | 99.95% |
| Clothing | 278,677 | 39,387 | 23,033 | 99.97% |

setting [4, 23, 28], we optimize the prediction model by minimizing the Bayesian personalized ranking loss as follows:

$$\mathcal{L}_{\text{BPR}} = - \sum_{(u,i^+,i^-) \in \mathcal{R}} \log \sigma \left( \hat{r}_{u,i^+} - \hat{r}_{u,i^-} \right), \qquad (17)$$

where $\mathcal{R} = \{(u, i^+, i^-) | r_{u,i^+} = 1, r_{u,i^-} = 0, r_{u,i^+} \in \mathbf{R}, r_{u,i^-} \in \mathbf{R}\}$. $\hat{r}_{u,i^+} = \hat{e}_u \cdot \hat{e}_{i^+}^{\text{T}}$ refers to the interaction score of a positive sample, where $\hat{e}_u$ and $\hat{e}_{i^+}$ are user representation and item representation derived from embedding matrix $E = E_{ui} + \tilde{E}_{ui}$.

Finally, we optimize the whole model by minimizing the following loss function:

$$\mathcal{L} = \mathcal{L}_{\text{BPR}} + \lambda_{ui}\mathcal{L}_{ui} + \lambda_{ii}\mathcal{L}_{ii}, \qquad (18)$$

where $\lambda_{ui}$ and $\lambda_{ii}$ are hyper-parameters that control the contribution of two contrastive learning modules.

In the model inference, the interaction score between user $u$ and item $i$ is computed as follows:

$$preds(u,i) = \hat{e}_u \cdot \hat{e}_i^{\text{T}}, \qquad (19)$$

where $\hat{e}_u$ and $\hat{e}_i$ are user representation and item representation derived from embedding matrix $E = E_{ui} + \tilde{E}_{ui}$.

## 4 EXPERIMENTS

In this section, we conduct extensive experiments on three public datasets to study the following research questions:

**RQ1:** Does the SOIL improve recommendation performance?

**RQ2:** What is the contribution of the three graphs and contrastive learning modules in improving performance?

**RQ3:** What is the effect of different hyper-parameters on the model performance?

**RQ4:** Does the second-order interest identify the items with which a user would like to interact?

**RQ5:** Do contrastive learning modules improve consistency between user-item embeddings and multimodal embeddings?

### 4.1 Datasets

To evaluate our method, we conduct extensive experiments on three widely used Amazon review datasets [5, 13]. Following [4, 23, 28], we choose the following categories: (a) Baby, (b) Sports and Outdoors, and (c) Clothing, Shoes and Jewelry. They are commonly referred to as Baby, Sports, and Clothing. All three datasets contain visual images and textual descriptions of the items. Following the widely used setting [4, 23, 26, 28], we use pre-extracted features: 4,096-dimensional for visual data and 384-dimensional for textual data. In our work, all items and users are filtered using the 5-core setting, and the statistics of datasets are reported in Table 1.

### 4.2 Experimental Protocols

**Evaluation Metric.** Following [4, 24, 29], we adopt the all-ranking protocol with two widely used metrics to evaluate the performance

Anonymous Authors

**Table 2: The performance comparison of different models. All results are reported in terms of Recall (R@K) and NDCG (N@K). The best and second results are marked with Bold and Underline. The improvement (*Improv.*) is calculated by comparing it with the best baseline on each dataset.**

| Dataset | Metric | BPR UAI'09 | LightGCN SIGIR'20 | VBPR AAAI'16 | LATTICE MM'21 | SLMRec TMM'22 | BM3 WWW'23 | FREEDOM MM'23 | MGCN MM'23 | LGMRec AAAI'24 | SIOL Ours | *Improv.* |
|---------|--------|-----|----------|------|---------|--------|-----|---------|------|--------|------|---------|
| Baby    | R@10   | 0.0357 | 0.0479 | 0.0423 | 0.0536 | 0.0540 | 0.0564 | 0.0627 | 0.0620 | 0.0644 | **0.0680** | 5.59% |
|         | R@20   | 0.0575 | 0.0754 | 0.0663 | 0.0858 | 0.0810 | 0.0883 | 0.0992 | 0.0964 | 0.1002 | **0.1028** | 2.59% |
|         | N@10   | 0.0192 | 0.0257 | 0.0223 | 0.0287 | 0.0285 | 0.0301 | 0.0330 | 0.0339 | 0.0349 | **0.0365** | 4.58% |
|         | N@20   | 0.0249 | 0.0328 | 0.0284 | 0.0370 | 0.0357 | 0.0383 | 0.0424 | 0.0427 | 0.0440 | **0.0454** | 3.18% |
| Sports  | R@10   | 0.0432 | 0.0569 | 0.0558 | 0.0618 | 0.0676 | 0.0656 | 0.0717 | 0.0729 | 0.0720 | **0.0786** | 7.82% |
|         | R@20   | 0.0653 | 0.0864 | 0.0856 | 0.0950 | 0.1017 | 0.0980 | 0.1089 | 0.1106 | 0.1068 | **0.1155** | 4.43% |
|         | N@10   | 0.0241 | 0.0311 | 0.0307 | 0.0337 | 0.0374 | 0.0355 | 0.0385 | 0.0397 | 0.0390 | **0.0435** | 9.57% |
|         | N@20   | 0.0298 | 0.0387 | 0.0384 | 0.0423 | 0.0462 | 0.0438 | 0.0481 | 0.0496 | 0.0480 | **0.0530** | 6.85% |
| Clothing| R@10   | 0.0206 | 0.0361 | 0.0281 | 0.0459 | 0.0452 | 0.0450 | 0.0629 | 0.0641 | 0.0555 | **0.0687** | 7.18% |
|         | R@20   | 0.0303 | 0.0544 | 0.0415 | 0.0702 | 0.0675 | 0.0669 | 0.0941 | 0.0945 | 0.0828 | **0.0998** | 5.60% |
|         | N@10   | 0.0114 | 0.0197 | 0.0158 | 0.0253 | 0.0247 | 0.0243 | 0.0341 | 0.0347 | 0.0302 | **0.0377** | 8.64% |
|         | N@20   | 0.0138 | 0.0243 | 0.0192 | 0.0306 | 0.0303 | 0.0295 | 0.0420 | 0.0428 | 0.0371 | **0.0456** | 6.54% |

of top-$K$ recommendations: Recall (Recall@$K$) and Normalized Discounted Cumulative Gain (NDCG@$K$). We report the average performance of all users with $K$ set to 10 and 20. In our research, we randomly divided the historical interactions of each user into training, validation, and testing sets with ratios of 8:1:1.

**Implementation Details.** In our work, we implement all compared methods and the proposed method with the MMRec framework [27]. For a fair comparison, we follow the experimental settings and pre-extracted visual and textual features in the MMRec framework. We use Recall@20 on the validation set as the indicator for early stopping. The embedding size for both users and items is set to 64 for all models. The number of GCN layers for the user-item graph is set to 2, and for item-item graphs, it is set to 1. We initialize all embedding parameters with the Xavier method [3], and optimize all models with the Adam optimizer [8]. Our method is implemented by Pytorch 1.12 and is trained on a single NVIDIA RTX 2080TI. Hyper-parameters $\lambda_{ui}$ and $\lambda_{ii}$ are chosen from $\{0.001, 0.01, 0.1, 1.0\}$. The $k$ for the $k$NN sparsification in Eq.(7) and Eq.(9) are chosen from $\{5, 10, 15, 20\}$. The $k$ for top-$k$ second-order interest in Eq.(2) is chosen from 1 to 10. **The code will be released on our website, and we also provide the model structure and the test code at the anonymous website (URL: https://anonymous.4open.science/r/soil-B1C3) for validation during the review phase.**

**Compared Methods.** In our work, we validate our approach by comparing it with the following traditional methods and multimodal recommendation methods:

(1) BPR [14] is a traditional method presenting a Bayesian personalized ranking optimization criterion to optimize the ranking models directly.
(2) LightGCN [7] is a traditional method that simplifies the GCN to make it more appropriate for recommendation.
(3) VBPR [6] is a multimodal method that incorporates visual signals into the collaborative filtering model and optimizes the model with Bayesian personalized ranking loss.
(4) LATTICE [24] is a multimodal method to construct item-item relationship graphs from multimodal features directly.

(5) SLMRec [16] is a multimodal method that performs data augmentation on multimodal features and deploys contrastive learning modules to optimize the model.
(6) BM3 [29] is a self-supervised multimodal recommendation method that reconstructs the user-item interaction graph and aligns multimodal features.
(7) FREEDOM [28] is a multimodal denoising method that freezes the item-item graph in LATTICE [24] and denoises the user-item graph to improve recommendation performance.
(8) MGCN [23] is a multimodal approach that reduces the effect of modal noise by treating user-item interactions and modal features as different views and performing graph convolutional learning in each of them separately.
(9) LGMRec [4] is a multimodal approach that learns representations from local and global graphs to model local and global user interest jointly.

## 4.3 Experimental Results

**RQ1: The ability to improve recommendation performance.** To evaluate the effect of our approach, we perform extensive experiments on three datasets and compare the results with several multimodal methods. We report the results of our method and other multimodal methods in Table 2. According to Table 2, our method is able to surpass the previous methods by a wide margin in all evaluation metrics. From the results, our method outperforms previous state-of-the-art methods by 5.59%, 7.82%, and 7.18% on Baby, Sports, and Clothing in terms of Recall@10. Similarly, our method is able to surpass previous state-of-the-art methods by 4.58%, 9.57%, and 8.64% on Baby, Sports, and Clothing in terms of NDCG@10. For the results of Recall@20, our method is able to achieve the improvement of 2.59%, 4.43%, and 5.60% compared to previous state-of-the-art methods on Baby, Sports, and Clothing. Furthermore, our method surpasses other methods by 3.18%, 6.85%, and 6.54% on Baby, Sports, and Clothing in terms of NDCG@20.

Among all the compared methods, MGCN has a similar network structure to our approach. According to the comparison in Table 2, our method is able to achieve the improvement of 9.68%, 7.82%, and

**Table 3: Results of the ablation study. SOI is short for the second-order interaction graph without first-order interest. IA is short for the interest-aware graph without the similarity graph. The ✓ indicates that the corresponding component is included.**

| Variants | | | | Baby | | | | Sports | | | | Clothing | | | |
| --- | --- | --- | --- | --- | --- | --- | --- | --- | --- | --- | --- | --- | --- | --- | --- |
| SOI | IA | $\mathcal{L}_{ui}$ | $\mathcal{L}_{ii}$ | R@10 | R@20 | N@10 | N@20 | R@10 | R@20 | N@10 | N@20 | R@10 | R@20 | N@10 | N@20 |
| | | | | 0.0556 | 0.0884 | 0.0303 | 0.0387 | 0.0650 | 0.0996 | 0.0354 | 0.0443 | 0.0509 | 0.0768 | 0.0279 | 0.0344 |
| | | ✓ | ✓ | 0.0634 | 0.0984 | 0.0351 | 0.0441 | 0.0722 | 0.1071 | 0.0400 | 0.0490 | 0.0665 | 0.0976 | 0.0364 | 0.0442 |
| ✓ | ✓ | | | 0.0609 | 0.0962 | 0.0327 | 0.0417 | 0.0667 | 0.1008 | 0.0361 | 0.0449 | 0.0597 | 0.0860 | 0.0324 | 0.0391 |
| ✓ | ✓ | ✓ | | 0.0642 | 0.1002 | 0.0348 | 0.0440 | 0.0722 | 0.1092 | 0.0400 | 0.0495 | 0.0602 | 0.0880 | 0.0330 | 0.0401 |
| ✓ | | ✓ | ✓ | 0.0658 | 0.0984 | 0.0357 | 0.0441 | 0.0749 | 0.1112 | 0.0410 | 0.0504 | 0.0675 | 0.0989 | 0.0369 | 0.0448 |
| | ✓ | ✓ | ✓ | 0.0674 | 0.1011 | 0.0367 | 0.0453 | 0.0759 | 0.1139 | 0.0413 | 0.0511 | 0.0680 | 0.0997 | 0.0377 | 0.0457 |
| ✓ | ✓ | ✓ | ✓ | 0.0680 | 0.1028 | 0.0365 | 0.0454 | 0.0786 | 0.1155 | 0.0435 | 0.0530 | 0.0687 | 0.0998 | 0.0377 | 0.0456 |

7.18% when compared with MGCN on Baby, Sports, and Clothing in terms of Recall@10. In addition, our method is able to outperform MGCN by 7.67%, 9.57%, and 8.64% on Baby, Sports, and Clothing in terms of NDCG@10. From Table 2, we have an observation that both our method and MGCN are able to outperform other methods. This observation indicates that the construction of separate graphs for historical user-item interactions and different modalities is beneficial for improving the performance of multimodal recommendations. Based on the comparison between traditional collaborative filtering methods and multimodal methods, we have another observation that most multimodal recommendation methods are able to exceed traditional collaborative filtering methods by a large margin in all evaluation metrics. This observation verifies that incorporating multimodal information into traditional models can significantly improve recommendation performance.

## 4.4 Modal Analysis

**RQ2: The contribution of different components.** To validate the contribution of each component in our work, we perform the ablation study on all evaluation datasets and report the experimental results in Table 3. In the ablation study, we focus on the effects of the second-order interaction graph, the interest-aware graph, the user-item contrastive learning module, and the item-item contrastive learning module in our framework. The first line in Table 3 indicates the baseline model without the above four components. From Table 3, our method is able to surpass the baseline model by 22.3%, 20.9%, and 35.0% on Baby, Sports, and Clothing in terms of Recall@10. Furthermore, our method is able to achieve the improvement of 20.4%, 22.9%, and 35.1% on Baby, Sports, and Clothing in terms of NDCG@10.

From Table 3, the second-order interaction graph and interest-aware graph are able to achieve the improvement of 9.53%, 2.6% and 17.2% compared to the baseline model on Baby, Sports, and Clothing in terms of Recall@10. The two contrastive learning modules are able to surpass the baseline model by 14.0%, 11.0%, and 30.6% on Baby, Sports, and Clothing in terms of Recall@10. From the last four lines in Table 3, both the interest-aware graph and the second-order interest graph contribute significantly to improving the performance of the recommender systems. This observation verifies that integrating second-order interest into the item-item graph is effective in improving recommendation performance. The comparison of all the results in Table 3 suggests that the primary contributors to performance improvement are the two contrastive

learning modules, while the second-order interaction graph and interest-aware graphs are secondary contributors.

**RQ3: The effect of hyper-parameters in our approach.**
**Effect of hyper-parameters $\lambda_{ui}$ and $\lambda_{ii}$.** The hyper-parameter $\lambda_{ui}$ controls the contribution of the loss function $\mathcal{L}_{ui}$ that is utilized to optimize the contrastive learning module of the user-item level. The hyper-parameter $\lambda_{ii}$ controls the contribution of the loss function $\mathcal{L}_{ii}$, which is utilized to optimize the contrastive learning module at the item level. We conduct experiments with different combinations of the two hyper-parameters and report the results in Figure 4(a). From the results, the performance of the model first increases and then gradually declines when $\lambda_{ii}$ increases. For the hyper-parameter $\lambda_{ui}$, the performance of the model shows a similar trend when $\lambda_{ui}$ increases. As $\lambda_{ui}$ decreases, the performance of the model decreases faster as $\lambda_{ii}$ increases. Our method is prone to achieve better prediction performance with smaller $\lambda_{ui}$ and $\lambda_{ii}$.

**Effect of $k$ in interest-aware item-item graphs.** In our work, we adopt different $k$ values in the top-$k$ operations of Eq.(7) and Eq.(9). The $k$ for interest in Eq.(7) controls the number of samples when selecting interest-aware samples. The $k$ for similarity in Eq.(9) controls the number of samples in selecting similar samples for each item. We conduct experiments with different combinations of the two $k$s and report the results in Figure 4(b). Note that the results are reported with all other parameters fixed and without tuning. From the results, the performance on Baby slowly grows up when $k$ for similarity increases, while the performance on Clothing fluctuates when $k$ for similarity increases. According to the results of $k$ for interest, the performance on Baby first grows up and then slowly decreases when $k$ increases. In our work, $k$ for similarity is set to 10 for all datasets. The $k$ for interest is set to 10 for Baby and 5 for both Sports and Clothing.

**Effect of $k$ when selecting second-order interest.** In our work, we adopt different $k$ values in the top-$k$ operations of Eq.(2). The $k$ in Eq.(2) controls the number of samples when selecting second-order interest samples. We conduct experiments on Baby and Sports and report the results in Figure 5. Note that the results are reported with all other parameters fixed and without tuning. From the results of Recall@10 in Figure 5, the performance on Sports slowly grows as $k$ increases, while the performance on Baby first increases and then slowly decreases as $k$ increases. The results of NDCG@10 show similar trends to the results of Reacall@10. According to the ablation study in Table 3, the second-order interaction graph is undoubtedly capable of improving the model

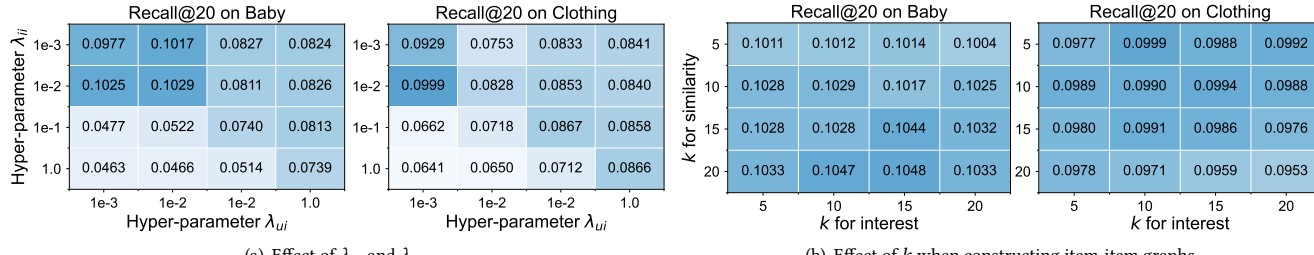

(a) Effect of $\lambda_{ii}$ and $\lambda_{ui}$

(b) Effect of $k$ when constructing item-item graphs

**Figure 4: Parameter sensitivity analysis on Baby and Clothing.**

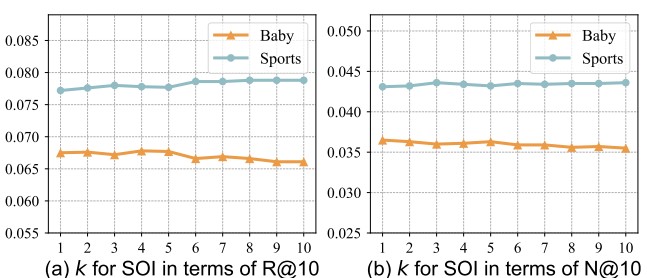

(a) $k$ for SOI in terms of R@10    (b) $k$ for SOI in terms of N@10

**Figure 5: Effect of $k$ in selecting second-order samples. SOI is short for the second-order interest.**

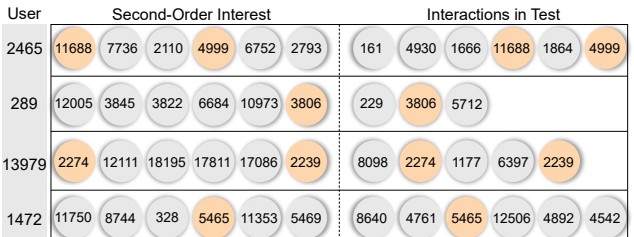

**Figure 6: Case study of the second-order interest on Sports. The orange circles indicate that the item is present in both the second-order interest and test set interactions, i.e., the second-order interest is valid.**

performance. The observation in Figure 4 indicates that the $k$ for selecting second-order interest depends on the dataset and varies widely across datasets. In our work, to achieve better performance, the $k$ for second-order interest is set to 2 for Baby, 6 for Sports and 3 for Clothing.

**RQ4: The effectiveness of second-order interest.** In our work, we propose to exploit the second-order interest of each user to retrieve the items in the candidate list. To qualitatively verify that the proposed second-order interest is practical in selecting potential interaction candidates, we compare the items on the second-order interest list with those on the interaction list of the test set and present the comparison results in Figure 6. Note that second-order interest is generated with the training data. From the results, our method is able to select two potential interaction candidates for user 2465 and user 13979. For users 289 and 1472, our method is able to select one potential interaction candidate. This observation verifies that the proposed second-order interest is able to retrieve potential interaction candidates.

**RQ5: The effect of contrastive learning modules.** In our work, we design contrastive learning modules at both the user-item

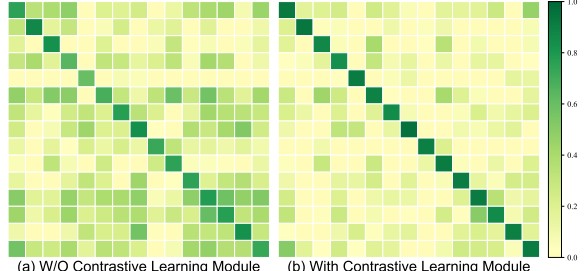

(a) W/O Contrastive Learning Module    (b) With Contrastive Learning Module

**Figure 7: Comparison of cosine similarity between user-item embeddings and multimodal embeddings. High diagonal similarity and low similarity in other areas indicate consistency between user-item embeddings and multimodal embeddings. W/O is short for without.**

and item-item levels to maintain user interest consistency between user-item embeddings and multimodal embeddings. To verify the ability to enhance the consistency of interest, we randomly selected user-item embeddings and multimodal embeddings and calculated the cosine similarity between them. The results are illustrated in Figure 7. In Figure 7, the vertical axis represents user-item embeddings, and the horizontal axis represents multimodal embeddings. The results indicate that in the method employing comparative learning modules, the multimodal embeddings and user-item embeddings of the same interaction are more closely aligned, whereas the similarity among different interactions is reduced. This observation verifies that contrastive learning modules in our work are able to enhance the feature consistency between user-item embeddings and multimodal embeddings.

## 5 CONCLUSION

This work presents a Second-Order Interest Learning framework to exploit the second-order interest from unrecorded candidate items. In this framework, we exploit candidate items from the perspective of constructing interest-aware graphs and self-supervised optimization. To construct interest-aware graphs, we first design an augmented user-item interaction graph that contains both first-order and second-order interest, and then we integrate the user interest with multimodal features to construct the interest-aware item-item graph for each modality. In self-supervised optimization, we design a user-item level contrastive learning module and an item-item level contrastive learning module to enhance the user interest between user-item embeddings and multimodal embeddings. Extensive experiments verify that our method is capable of significantly improving the recommendation performance.

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
