# OpenReview forum: "SOIL: Contrastive Second-Order Interest Learning for Multimodal Recommendation"
_acmmm.org/ACMMM/2024/Conference — MM2024 Poster_

### Official Review · Reviewer_rkt8 · 2024-05-23

**Rating:** 4
**Confidence:** 3

**Summary:**

To exploit more accurate user interests, the paper proposes a second-order interest learning framework. In this work, the author denotes items on the interaction list as first-order interests and items on the candidate list as second-order interests. These interests are separately used to construct interest graphs and learn structural information within the graphs via LightGCN. Furthermore, the paper constructs interest-aware item-item graphs using a feature similarity measure, which enhances the relationship between multimodal features and user interests. Finally, pairwise contrastive learning is employed to align the information between ID embeddings and multimodal embeddings.

**Strengths:**

1. The paper highlights the importance of second-order interest in completing user preferences and proposes a structured method by calculating the attractiveness score.
2. The paper's motivation is well-articulated, with the authors clearly outlining the three problems they aim to solve. This clarity strengthens the rationale behind the proposed methods.
3. The paper conducts extensive experiments and ablation studies on three datasets, verifying the effectiveness of the proposed method.

**Limitations:**

1. There is an ambiguous description of the interest-aware graph construction process in Section 3.3. Specifically, the author first calculates the similarity sum for each modality feature via Equation 6, but then introduces the top-𝑘 function to select k items to represent user interest.
2. There is a redundant formulation of Equation 14.
3. There is a misleading explanation of Equation 16. The paper states that it "deploys a contrastive learning module on embeddings of the user-item graph and item-item graph," but the formulation does not reflect the purpose as described by the authors.

**Suitability:**

2

---

### Official Review · Reviewer_ht2s · 2024-05-24

**Rating:** 4
**Confidence:** 2

**Summary:**

Conventional multimodal recommender systems learn user interests from user-item interaction graphs, but they only capture first-order interests and ignore suboptimal items. To address this problem, the authors propose a Second-Order Interest Learning (SOIL) framework that extracts second-order interests from unrecorded suboptimal items. The framework constructs user-item and interest-aware item-item graphs for visual and textual modalities using multimodal feature similarity. Graph convolutional networks are used to learn user and item representations from these graphs. Contrastive learning modules further enhance the exploitation of both first- and second-order interests. Evaluated on three real-world datasets, the SOIL framework significantly improves prediction performance.

**Strengths:**

The authors have made the code and dataset publicly available, which increases the credibility of the paper's results.
The experimental results seems convincing.

**Limitations:**

a)	At the end of the second paragraph of the Introduction, it is mentioned that "Experimental results in recent reports have demonstrated that learning on interaction-based graphs improves prediction accuracy significantly". Suggest adding a citation after 'reports'.
b)	The article frequently uses the simple past tense or present perfect tense. It is recommended to use the simple present tense.
c)	It is recommended that the author add the differences and connections between the method in this paper and the methods mentioned in the Related Work section.
d)	The formatting of symbols in the text is inconsistent. Symbols representing matrices should be uppercase, upright, and bold, while symbols representing vectors should be lowercase, upright, and bold.
e)	Please increase the clarity of Figure 1. It is recommended to enlarge the text in the image above the word 'Search'.

**Suitability:**

3

---

### Official Review · Reviewer_g2vd · 2024-05-27

**Rating:** 3
**Confidence:** 3

**Summary:**

This paper introduces a comprehensive framework designed to enhance the accuracy and relevance of recommendations in multimodal recommender systems. The core design of this work is introducing a concept called second-order interests, which are essentially potential user interests that are typically overlooked in conventional user-item interaction data because they do not result in actual interactions. This paper utilizes multimodal information to obtain this information and augment standard user-item graphs. It further employs contrastive learning to refine the user and item embeddings at both the user-item and item-item levels, thereby enhancing the consistency and relevance of the learned embeddings. The proposed methods are evaluated on three real-world datasets, showing improvements in recommendation accuracy over existing methods.

**Strengths:**

- Interesting idea to exploit second-order interest. Exploiting more useful information from limited feedback information is a key problem in recommender systems. Leveraging multimodal information is reasonable.

- The proposed method has been compared with many SOTA methods, and the performance is good.

**Limitations:**

- The novelty is somewhat limited, and the proposed solution seems straightforward. Both constructing an augmentation graph and contrastive learning are not new.

- The proposed solution may add extra burden in training and inference, which should be discussed and analyzed.

**Suitability:**

3

---

### Meta-Review · Area_Chair_JQ6S · 2024-07-09

**Recommendation:** Accept (Poster)
**Confidence:** 4

**Metareview:**

The reviewers agree that the paper is interesting and relevant to a large part of ACM Multimedia community. They further agree that the idea of utilizing what authors term as second-order interests instead of conventional historical interactions encoded in user-item matrix is interesting. However, they have identified certain drawbacks as well, such as the need for better describing the exact novelties of the proposed approach as well as the need for clarifying the equations in Section 3.3.

I believe that the paper should be accepted to the program of ACM Multimedia 2024. When preparing a revised version of the paper the authors should carefully address reviewer comments and proofread the paper for any grammar (e.g. an issue with the tenses was suggested by one of the reviewers) and typographical errors.